# Temporal assessment of entomological surveillance of *Trypanosoma cruzi* vectors in an endemic area of northeastern Brazil

George Harisson Felinto Sampaio[1]*, Andressa Noronha Barbosa da Silva[2], Christiane Carlos Araújo de Negreiros[2], Nathan Ravi Medeiros Honorato[3], Rand Randall Martins[2], Lúcia Maria Abrantes Aguiar[4], Letícia Mikardya Lima Sales[5], Carlos Ramon do Nascimento Brito[6], Paulo Marcos da Matta Guedes[6], Antonia Claudia Jácome da Câmara[2,6], Lúcia Maria da Cunha Galvão[1,2,3]

1 Programa de Pós-Graduação em Ciências da Saúde, Centro de Ciências da Saúde, Universidade Federal do Rio Grande do Norte, Natal, RN, Brasil, 2 Programa de Pós-Graduação em Ciências Farmacêuticas, Centro de Ciências da Saúde, Universidade Federal do Rio Grande do Norte, Natal, RN, Brasil, 3 Programa de Pós-Graduação em Parasitologia, Instituto de Ciências Biológicas, Universidade Federal de Minas Gerais, Belo Horizonte, MG, Brasil, 4 Secretaria de Estado da Saúde Pública do Rio Grande do Norte, Natal, RN, Brasil, 5 Curso de Graduação em Farmácia, Centro de Ciências da Saúde, Universidade Federal do Rio Grande do Norte, Natal, RN, Brasil, 6 Programa de Pós-Graduação em Biologia Parasitária, Centro de Biociências, Universidade Federal do Rio Grande do Norte, Natal, RN, Brasil

* felintosampaio@hotmail.com

**Data Availability Statement:** All relevant data are within the paper and its Supporting Information files.

## Abstract

Entomological surveillance is essential for the control of triatomines and the prevention of *Trypanosoma cruzi* infection in humans and domestic animals. Thus, the objective of this study was to evaluate entomological indicators and triatomine control during the period from 2005 to 2015 in an endemic area in the state of Rio Grande do Norte, Brazil. This observational and retrospective study was developed based on data analysis related to active entomological surveillance activities and chemical control of infested housing units (HU) in the Agreste mesoregion of the state of Rio Grande do Norte, Brazil, in the period between 2005 to 2015. The quantitative analysis of housing units surveyed for entomological indicators was performed by linear regression of random effects ($p < 0.05$). The effect of the number of HU surveyed on the entomological indicators was analyzed by fitting a linear random effects regression model and an increasing intradomiciliary colonization rate was significant. In the period evaluated 92,156 housing units were investigated and the presence of triatomines was reported in 4,639 (5.0%). A total of 4,653 specimens of triatomines were captured and the species recorded were *Triatoma pseudomaculata* ($n = 1,775$), *Triatoma brasiliensis* ($n = 1,569$), *Rhodnius nasutus* ($n = 741$) and *Panstrongylus lutzi* ($n = 568$), with an index of natural infection by *T. cruzi* of 2.2%. Only 53.1% of the infested HU were subjected to chemical control. Moreover, there was a decrease in the total number of HU surveyed over time associated with an increase in the index of intradomiciliary colonization ($p = 0.004$). These data demonstrated that entomological surveillance and control of vectors in the Agreste mesoregion of the state has been discontinued, emphasizing the need for more effective public policies to effectively control the vectors, in order to avoid the exposure of humans and domestic animals to the risk of *T. cruzi* infection.

**Funding:** This work was supported by research grants from the Conselho Nacional de Desenvolvimento Científico e Tecnológico MCTI/CNPq/universal number 423966/2016-2 (ACJC), and CNPq fellows (LMCG) and Coordenação de Aperfeiçoamento de Pessoal de Nível Superior-CAPES fellows (ANBS, GHFS and NRMH). And Conselho Nacional de Desenvolvimento Científico e Tecnológico, 423966/2016-2, Antônia Cláudia Jácome Da Câmara. The funders had no role in study design, data collection and analysis, decision to publish, or preparation of the manuscript.

**Competing interests:** This work has not been submitted elsewhere for publication and will not be until we receive your reply regarding publication in your Journal. And also, if accepted, it will not be published elsewhere in the same form, in English or in any other language, including electronically without the written consent of the copyright-holder. This does not alter our adherence to PLOS ONE policies on sharing data and materials.

## Introduction

Chagas disease (ChD) is a neglected tropical disease caused by *Trypanosoma cruzi* Chagas, 1909, resulting in high morbidity in endemic countries and affecting mainly low- income communities, who usually live in rural areas [1, 2]. Approximately six million people are infected with *T. cruzi* in 21 Latin American countries, with 75 million individuals at risk of acquiring the infection [3].

The hematophagous insects known as triatomines (Hemiptera, Reduviidae, Triatominae) are the main means of transmission of the parasite *T. cruzi* in Latin America [1]. Currently, 154 triatomine species have been identified [4–8] and all of them are potential vectors of the parasite [9]. From an epidemiological point of view, the most important species in Brazil are *Panstrongylus megistus* Burmeister, 1835, *Triatoma brasiliensis* Neiva, 1911, *Triatoma sordida* Stål, 1859 and *Triatoma pseudomaculata* Corrêa and Espínola, 1964 due to the adaptation of these insects to peridomestic and intradomestic ecotopes, in addition to their vectorial capacity and vector competence [10].

The northeast region of Brazil has a wide distribution of native species of triatomines, such as *T. brasiliensis*, *T. pseudomaculata* and *Rhodnius nasutus* Stål, 1859 [11, 12], with different rates of natural infection by *T. cruzi*. Several studies have reported that *T. brasiliensis* and *T. pseudomaculata* are the most captured species in housing units (HU, including both intra- and peridomestic locations) in the states of Ceará [13–15], Paraíba [16, 17], Pernambuco [18], Bahia [19] and Rio Grande do Norte [16, 20, 21].

In the state of Rio Grande do Norte (RN), there are several sources of evidence that the transmission of *T. cruzi* is still active. Several autochthonous species of triatomines have been captured in HU in RN, such as *T. brasiliensis*, *T. pseudomaculata*, *Panstrongylus lutzi* Neiva and Pinto, 1923, *P. megistus*, *T. petrocchiae* Pinto and Barreto, 1925 and *R. nasutus* [20–23]. These species showed high rates of natural infection by *T. cruzi* and the ability to colonize different areas of the state, in the municipalities of the Central and West mesoregions [21, 24, 25]. In addition, an outbreak of oral transmission of *T. cruzi* (i.e., through human ingestion of food contaminated with parasite-infected triatomines) was identified in the municipality of Marcelino Vieira, located in the Agreste mesoregion, indicating the presence of infected triatomines in contact with humans, strongly suggesting active vector transmission in this mesoregion of the state [26]. Furthermore, last national seroprevalence survey to assess the control of ChD in Brazil, carried out in children aged 0 to 5 years living in rural areas, and whose mothers were seronegative, highlighted that 1 out of 1,750 samples analyzed from the state of RN was positive for anti-*T. cruzi* antibodies, suggesting probable vector transmission in the Agreste mesoregion of the state [27].

Entomological surveillance (ES) of *T. cruzi* vectors should be continuous where there is peri- and intradomestic invasion and colonization by triatomines [28]. These actions contribute to the control of vector transmission of *T. cruzi* through measures that involve educational programs for the human populations at risk, the capture of triatomines in HU and peridomestic ecotopes, and, principally, chemical control in areas infested by vectors infected by *T. cruzi* and their adjacent surroundings [1]. These measures prevent or reduce the possibility of contact between humans and infected triatomines, prevent colonization in the HU and/or eliminate existing colonies [29]. Entomological indicators are used to assess the risk of colonization of HU by either nymphs or adult triatomines, and their analysis allows selection of the HU that will be subjected to residual spraying [1]. The success of ES and the control of *T. cruzi* vectors is characterized by the elimination of triatomine colonies in intra- and peridomestic ecotopes, leading to a reduction in vector transmission and, consequently, better control of ChD transmission [30–32].

Therefore, the aim of this study was to evaluate entomological indicators and triatomine control and control measures between 2005 to 2015 in an endemic area for ChD in the state of Rio Grande do Norte, Brazil.

## Methods

### Study área

The state of RN is located in the northeast of Brazil, and is represented in Fig 1, with emphasis on the three maps, A, B and C, which correspond to the map of Brazil, the state of RN and the agreste mesoregion, respectively. The RN has a total population of approximately 450,000 individuals living in risk areas, of which 47,633 individuals live in areas considered high risk [33]. It is located in the northeast of Brazil, and has a total area of 52,810.7 km$^2$, corresponding 3.14% of the territory of the country, and is divided into 167 municipalities distributed throughout four mesoregions: the West, Central, Agreste and East [33]. The Agreste mesoregion (**Fig 1C**) was the focus of this study because it comprises 48.8% of the municipalities considered at high or medium risk of vector transmission of *T. cruzi*. It is located between the "Mata" and "Sertão" zones, with a semi-arid and dry climate, low rainfall and the Caatinga as its main biome. This mesoregion has a total of 43 municipalities, and according to the risk stratification for vector transmission of *T. cruzi*, 20.9% of these are considered at high risk, 27.9% at medium risk, and 51.1% at low risk, according to data from the Ministry of Health for the year 2012, provided by the State Department of Public Health of Rio Grande do Norte (SESAP-RN) (**S1 File**). The estimated population in this area was 452,558, with about 180,000

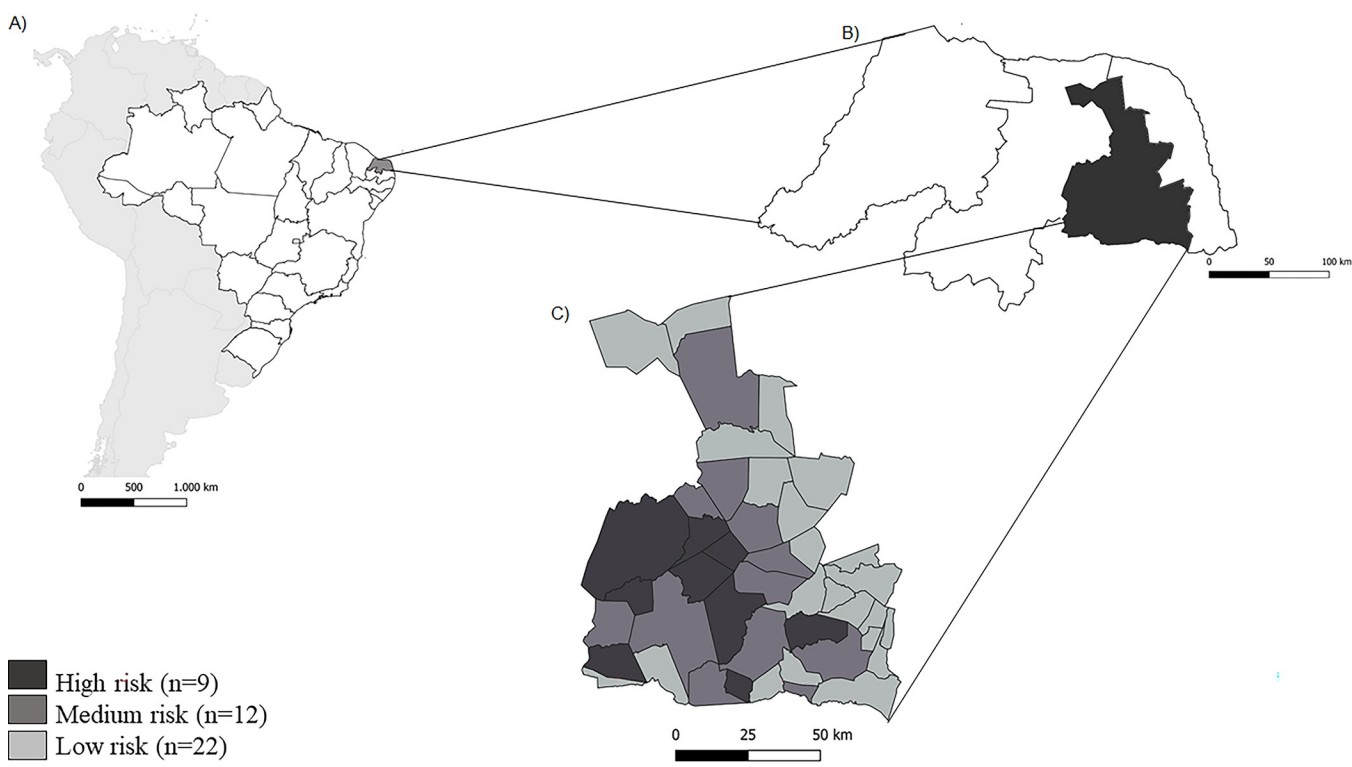

High risk (n=9)
Medium risk (n=12)
Low risk (n=22)

**Fig 1.** The state of RN is located in the northeast of Brazil, with emphasis on the three maps, A, B and C, which correspond to the map of Brazil, the state of RN and the agreste mesoregion, respectively.

residing in rural areas, which are considered the most exposed to vector transmission. According to the last census [34], the region has a Human Development Index (HDI) of 0.578.

## Entomological data

This was an observational and retrospective population-based study based on entomological data associated with the capture of triatomines in the HUs of the Agreste mesoregion from 2005 to 2015. The data were obtained from the SESAP-RN database, which manages the information collected by municipalities that carry out epidemiological surveillance activities. In this study, information regarding the occurrence of triatomine species, their developmental stage, capture environment (intra- or peridomestic) and the number of insects infected with *T. cruzi* were used. Triatomine captures were carried out manually in the HU by agents for the control of endemic diseases of the local health departments, without the use of insect dislodging substances. In each HU, two public health workers investigated the intradomestic environment as well as the peridomestic area and their surrounding annexes. The identification of triatomines was performed according to the identification key proposed by Lent & Wygodzinsky [35]. In order to determine the rate of natural infection by *T. cruzi*-like, a direct parasitological examination was performed by abdominal compression of the insects and the feces were diluted in saline solution and aliquots were examined between a slide and a coverslip, using optical microscopy at 400× magnification.

Maps were elaborated by the authors using QGIS software version 3.16.15 (<http://qgis.osgeo.org>) with UTF-8 encoding and Geocentric Reference System for the Americas (SIR-GAS 2000) data obtained from public database of Brazilian Institute of Geography and Statistics (IBGE), accessed in January 2022 at <https://portaldemapas.ibge.gov.br/portal.php#homepage>.

Data regarding the surveyed and positive HU were used to evaluate chemical control interventions and to calculate entomological indicators, according to criteria established by the World Health Organization [36] and the Pan American Health Organization [37]. The indicators used were: (i) infestation rate (number of infested HU × 100/number of surveyed HU); (ii) intradomestic infestation rate (number of infested intradomiciles × 100/number of surveyed intradomiciles); (iii) peridomestic infestation rate (number of infested peridomiciles × 100/number of surveyed peridomiciles); (iv) intradomestic colonization rate (number of HU with nymphs in the intradomicile × 100/number of HU with triatomines in the intradomicile) and (v) natural infection rate (number of triatomines infected by *T. cruzi* × 100/number of triatomines examined). The intradomicile corresponds to the interior of human dwellings and the peridomicile refers to the area around the dwelling which is affected by anthropic activities [38].

## Statistical analysis

Data were descriptively analyzed and presented as mean, standard deviation and relative and absolute frequencies. The variation between the percentage of investigated HU and the percentage of sprayed HU in each municipality was analyzed by simple linear regression (p<0.05). The effect of the number of HU surveyed on the entomological indicators was analyzed by fitting a linear random effects regression model. The entomological indicators were considered as the dependent variable, the period of investigation as the independent variable of fixed effect, and the quantity of surveyed HU as the random variable of the model. Correlation coefficients (β) were presented for each epidemiological indicator with their respective 95% confidence intervals (95%CI). *P* values less than 0.05 were considered significant. All analyzes were performed using Stata v15 software.

**Table 1. Profile of the housing units surveyed and their entomological indicators between the years 2005 to 2015.**

| Year | Housing units (HU) | | | | | Entomological indicators (%) | | | |
|---|---|---|---|---|---|---|---|---|---|
| | Surveyed | Positive | % | Sprayed | % | ICR | IR | PIR | IIR |
| 2005–2007 | 35.211 | 2.627 | 7,5 | 1.073 | 40,8 | 18,2 | 7,5 | 1,3 | 2,5 |
| 2008–2010 | 26.302 | 654 | 2,5 | 586 | 89,6 | 17,4 | 2,5 | 1,2 | 1,8 |
| 2011–2013 | 19.876 | 983 | 4,9 | 511 | 52,0 | 25,3 | 4,9 | 1,3 | 3,4 |
| 2014–2015 | 10.767 | 375 | 3,5 | 293 | 78,1 | 24,8 | 3,5 | 1,6 | 2,4 |
| **Total** | **92.156** | **4.639** | **5,0** | **2.463** | **53,1** | **21,4** | **5,0** | **1,3** | **2,6** |

ICR: intradomestic colonization rate; IR: infestation rate; PIR: peridomestic infestation rate, and IIR: intradomestic infestation rate.

## Results

An active search for triatomines (manual collection) was carried out between 2005 to 2015 in 92,156 HU, and these insects were found in 4,639 (5%) of them (**Table 1**). A total of 4,653 triatomines were captured in the HU of the Agreste mesoregion of RN, and the most commonly identified species were: *T. pseudomaculata* (*n* = 1,775), *T. brasiliensis* (*n* = 1,569), *R. nasutus* (*n* = 741) and *P. lutzi* (*n* = 568). *T. brasiliensis* and *T. pseudomaculata* were also the most common species in both intra- and peridomestic ecotopes. The rate of natural *T. cruzi* infection was 2.2%, with *T. brasiliensis* showing the highest infection rate (2.8%), followed by *T. pseudomaculata* (2.4%) (**Table 2**).

The data showed a decrease in the number of HU surveyed over time. In the period from 2005 to 2007, 35,211 HU were investigated with 7.5% of them infested. In the last period evaluated (2014 to 2015) a decrease of about three and half times was observed in the number of HU surveyed (10,767) while the percentage of triatomine-positive HU (3.5%) also decreased just over two-fold (**Table 1**). The analysis of ES per year showed that after 2012, even the municipalities considered at high risk of vector transmission control activities were discontinued, with coverage around 60 to 80% observed in the period from 2012 to 2015 (**S2 File**).

Linear regression analysis showed that the decrease in the number of triatomine-positive HU (p = 0.005) is directly associated with the decrease in the number of HU surveyed (p = 0.014) (**Fig 2**). This correlation between the decrease in ES services and the decrease in the detection of HU infested by triatomines was observed over time (**S3 File**).

Chemical control was performed in only 53.1% (2,463/4,639) of the triatomine-positive HU and, therefore, a total of 2,176 HU infested by triatomines were not subjected to vector control measures (**Table 1**). The insecticide spraying was carried out using a synthetic pyrethroid (alpha-cypermethrin SC 20%) at a concentration of 0.04 g of active ingredient and with

**Table 2. Natural infection rate by *Trypanosoma cruzi* in different species of triatomines according to their developmental stage and intra- or peridomestic locations.**

| Especies | Intradomicile | | | | Peridomicile | | | | Total | % | Natural infection rate (%) |
|---|---|---|---|---|---|---|---|---|---|---|---|
| | A | N | T | % | A | N | T | % | | | |
| *Triatoma brasiliensis* | 179 | 105 | 284 | 6.1 | 863 | 422 | 1,285 | 27.6 | 1,569 | 33.7 | 2.8 (44/1,569) |
| *Triatoma pseudomaculata* | 173 | 122 | 295 | 6.3 | 1,047 | 433 | 1,480 | 31.8 | 1,775 | 38.2 | 2.4 (35/1,478) |
| *Rhodnius nasutus* | 33 | 37 | 70 | 1.5 | 450 | 221 | 671 | 14.4 | 741 | 15.9 | 1.4 (10/607) |
| *Panstrongylus lutzi* | 0 | 2 | 2 | 0.1 | 449 | 177 | 566 | 12.2 | 568 | 12.2 | 0.2 (01/450) |
| **Total** | 385 | 266 | 651 | 14.0 | 2,809 | 1.193 | 4,002 | 86 | 4,653 | 100 | 2.2 (90/4,104) |

A: number of adults; N: number of nymphs; T: total number; %: percentagem.

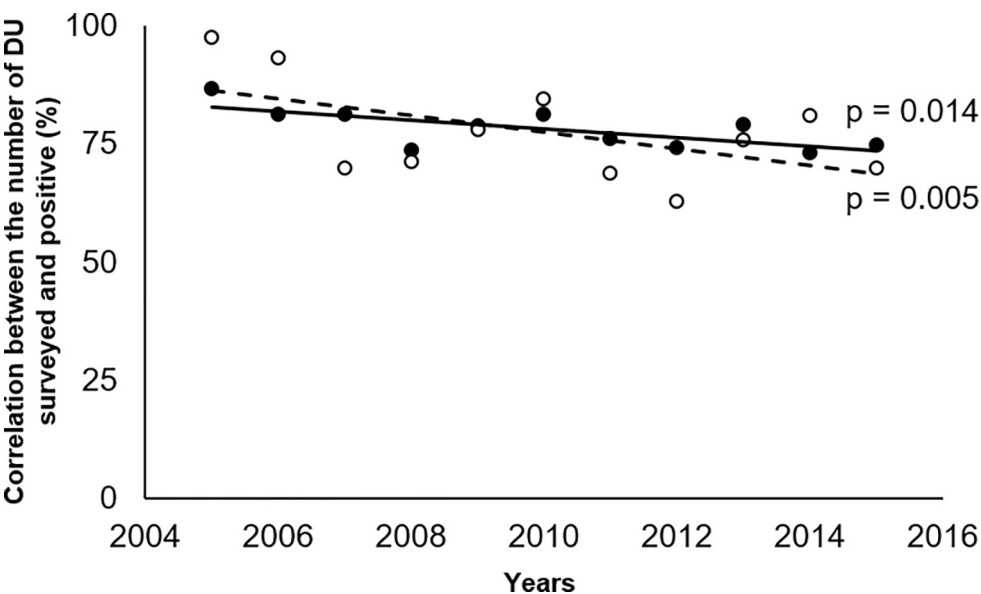

**Fig 2.** Comparison of the correlations between (i) the percentage of all housing units (HU) surveyed (solid line, black circles), or (ii) the percentage of the latter that were triatomine-positive (dashed line, white circles), and the duration of the study period.

residual action lasting six to twelve months. It is important to emphasize that chemical control in positive HU did not reach 100% effectiveness in any year. Furthermore, a number of municipalities showed discontinuity in ES services in all the years of the period of evaluation of the current study (**S4 File**). In addition to this epidemiological scenario, an increasing intradomiciliary colonization rate was also detected. This colonization varied from 17.4% to 25.3% over the years investigated (**S5 File**). The infestation rate was higher in intradomestic (2.6%) when compared to peridomestic locations (1.3%) (**Table 1**). Thus, the use of the number of HU surveyed as an adjustment variable in the multivariate regression model showed that the intra-household colonization index significantly increased over the years (β coefficient = 0.015; $p = 0.004$) (**Fig 3**).

## Discussion

The entomological surveillance and control of the triatomine vectors of *T. cruzi* was evaluated, with an emphasis on the chemical control of triatomines in the Agreste mesoregion of RN, a state in northeastern Brazil, where most individuals reside in rural areas, which are considered vulnerable to ChD. The main findings of our study were that: (i) residual spraying was performed in only 53.1% of all HU infested by triatomines, and (ii) a decrease in the number of surveyed and triatomine-positive HU over time contrasted with an increase in intradomestic colonization, especially by native triatomine species, such as *T. brasiliensis* and *T. pseudomaculata*. This epidemiological scenario suggests the existence of an active cycle of vector transmission of *T. cruzi*, likely as a result of the discontinuation of public health interventions and with the possibility of serious health consequences to the local human populations [1].

The difficulty in eliminating the etiological agent of ChD is mainly due to the lack of vaccine, and the complexity of the transmission cycle, which has several parasite reservoirs. Thus, surveillance activities that prevent vector contact with humans is the most effective way to prevent transmission and promote infection control [1, 29]. The risk of transmission of *T. cruzi* persists due to the existence of autochthonous triatomine species with colonization potential

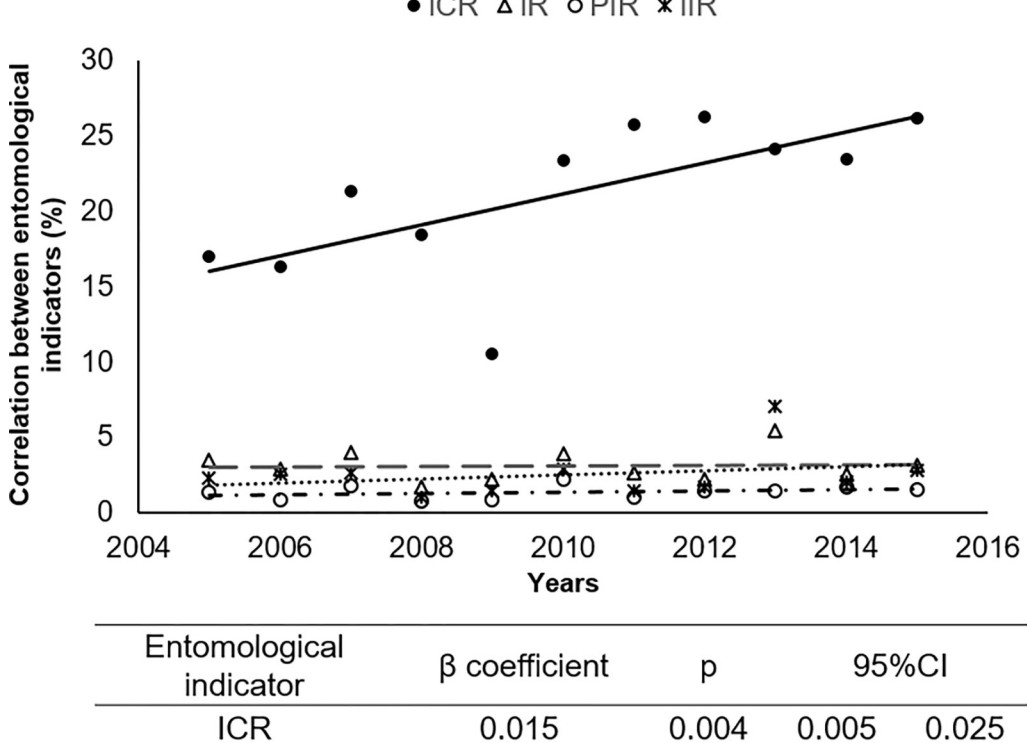

Fig 3. Linear random effects regression model describing the variation of epidemiological indicators. (ICR: intradomestic colonization rate; IR: infestation rate; PIR: peridomestic infestation rate, and IIR: intradomestic infestation rate between the years 2005 to 2015 adjusted according to the quantity of housing units investigat).

and therefore, the actions of ES must consider not only the biological characteristics of the vectors, but also the environmental influence on the occurrence of behavioral changes, requiring the development of different control strategies according to the potential risk for vector transmission [39]. The prevention of ChD is directly related to the form of parasite transmission, and the measures that prevent the transmission of *T. cruzi* to susceptible humans [40].

The analysis of entomological indicators such as the intra- and peridomiciliary infestation rate and intradomiciliary colonization suggests that the risk of ChD among the people living in the study area increased during the period investigated. Intrahousehold colonization increased significantly over time, ranging from 17.4% to 25.3% when the number of HU surveyed was suspected as an adjustment variable in the multivariate regression analysis. The increasing value of this indicator provides an important correlation between the presence of the insect vector in the nymphal stage with the intradomestic ecotope, indicating possible reinvasion, reinfestation and even domiciliation in the HU [1].

Another important finding was the detection of the highest infestation rates in the intradomestic ecotope throughout the entire study period. These entomological indicators highlight the proximity between humans and triatomines, and identify possible active cycles of vector transmission of *T. cruzi* [25, 41–43]. Intra- and peridomestic infestations have also been identified in other states in northeastern Brazil, showing the capacity for invasion and/or

colonization of HU by different species of triatomines, mainly *T. brasiliensis* and *T. pseudomaculata* [42, 44]. The northeastern semi-arid region stands out in the national context for its distinct ecoepidemiological aspects, demonstrating great diversity and wide dispersion of triatomines, which requires permanent and effective strategies to control these vectors [41].

*Triatoma brasiliensis* and *T. pseudomaculata* have been frequently identified in the state of RN, and the infestation and colonization of intra- and peridomestic locations by these species, considered important vectors of *T. cruzi* in Brazil, have been reported [16, 20, 21, 23, 24, 29, 45, 46]. *Triatoma brasiliensis* has been identified with high rates of natural infection by *T. cruzi* in other areas of RN, with positivity rates ranging from 19.2% to 50.0% [21], reaching up to 79.0% when molecular techniques were used for parasite identification [20]. In addition, cultural habits, such as the presence of breeding sites for domestic animals, or piles of tiles, bundles of straw, or stacks of bricks and wood close to the HU are characteristics identified in the studied area that facilitate triatomine invasion and colonization [47]. These factors may favor the emergence of new triatomine foci, even in urban areas with precarious housing conditions [48].

In our study, the overall observed infection rate of triatomines by *T. cruzi* was 2.2%, when using the direct parasitological method, which is used in surveillance programs in Brazil, and in general has a low sensitivity and reproducibility, but a specificity of 100% for Trypanosomatidae [49]. However, the actual infection rate could be higher if more sensitive detection methods were used. A study developed in another mesoregion of the state of RN, using PCR for diagnosis, showed natural infection rates of up to approximately 100% in *T. brasiliensis* [20].

The existence of significant problems in the execution of entomological surveillance activities in the evaluated area, especially the chemical control of triatomines, did not allow the goal, recommended by the Brazilian Ministry of Health (MH), to residually spray all HU infested with infected vectors, in addition to adjacent surrounding areas. The infested areas without residual spraying become a favorable habitat for the proliferation of triatomines, with abundant vertebrate hosts for blood-feeding because of the availability of humans and various species of domestic and peridomestic animals, in addition to resting sites and hiding places in the HU. Thus, spraying the adjacent surrounding areas prevents large-scale spatial displacement from infested areas of triatomines in search of bloodmeals [1]. Recent data have shown that the efficacy of chemical control in peri- and intradomestic ecotopes is approximately six months, with a significant decrease after 14 months of application. These results are not in accordance with MH guidelines for triatomine control, which recommends 24-month protection after residual spraying of treated areas. The consequence of the increase in the triatomine population is the greater exposure of individuals residing in these municipalities, and in nearby areas to where the insects may migrate in search of food, enabling the colonization of these adjacent areas and possibly establishing new transmission cycles [29, 50]. In addition, the triatomine species identified in the studied area have a wide geographic distribution, high rates of infection by *T. cruzi*, several vertebrate bloodmeal sources and are considered autochthonous, with the ability to colonize intra- and peridomestic ecotopes. This behavioral pattern of vectors favors invasion, reinfestation and colonization, making its control even more complex [1, 15, 41, 51–53].

Vector control strategies should focus on continuous ES in endemic areas, allowing the monitoring of HU and thus ensuring the sustainability of control interventions and early detection of triatomines in intra- and peridomestic locations. Chemical control should be carried out by spraying of insecticides, using chemical substances with residual action, both inside and in the annexes of all infested HU [1, 29, 54]. The vector control of triatomines depends on an effective chemical treatment that must be carried out systematically in all infested HU using adequate techniques and repeated application at regular intervals [55], otherwise there is a risk of any surviving insects remaining to colonize the HU [56]. Although triatomines are

susceptible to pyrethroids [57, 58] some species may be resistant to insecticides, especially *T. brasiliensis*, thus requiring more efficient and sustainable measures to control these vectors [56].

Interventions such as improving housing conditions and organizing the peridomicile with elimination of debris, stone fences and animal styes [1, 54], as well as keeping dogs, cats, rodents and birds out of human sleeping areas, also facilitate infection control [59]. The main objective of ChD vector control programs is to eradicate domestic colonies of triatomines and maintain continuous surveillance of *T. cruzi* vectors. In this sense, the WHO guidelines guide that active search in high and medium risk municipalities should be carried out in all infested HU (and in neighboring locations up to 1 km away) in which the presence of *T. cruzi* infected triatomines have been reported previously. Other sites should be selected and randomly included to assess infestation, using active search for triatomines. In municipalities classified as low-risk for *T. cruzi* transmission, ES is carried out passively, with the population participating in triatomine notification, followed by active ES [1, 29].

The decentralization of entomological control measures for mosquitoes, sand flies and triatomines by the Brazilian government began in 1999, transferring responsibility to local states and municipalities [60]. Concomitantly, the last few decades were marked by the occurrence of several arboviral epidemics (Dengue, Chikungunya and Zika viruses) in large Brazilian urban centers [61] and by the official declaration of the interruption of vector transmission of *T. cruzi* by *Triatoma infestans* Klug, 1834, in 2006, which contributed to the false belief that the problem of ChD in Brazil had been solved [29]. These factors had a negative impact on the control of triatomines, mainly due to local problems of management, and the re-allocation of resources to combat arboviruses, disproportionately affecting less- densely populated rural areas in northeastern Brazil [29, 62].

On the other hand, the effectiveness of ES in the state of São Paulo, where activities have taken place continuously, meant monitoring ensured the sustainability of early detection of triatomines and their control [63]. Successful public health policies in eliminating *T. cruzi* vectors have also been demonstrated in other countries such as Venezuela [31, 32], Ecuador [30], Uruguay and Argentina [64]. The lack of investment in ES activities in endemic areas leads to peaks in new cases of vector transmission, as observed in indigenous communities in Colombia [65] and Bolivia [66], which had a 60-fold higher incidence of *T. cruzi* infection when compared to regions with ongoing surveillance and control interventions. In Argentina, failure of ES in areas infested with *T. infestans* resulted in population recovery of these insects 2 to 3 years after discontinued spraying of insecticides [67].

The control of the vectors of *T. cruzi* in endemic areas must be carried out through active surveillance and public participation, undertaken simultaneously. Competent health agencies must install triatomine information posts in order to train the local population to identify the vector in HU and to notify the local entomological teams. Previously called "passive surveillance", these new approaches with public participation and continuing education should sensitize the community in the control of triatomines according to the reality of each municipality. These activities are extremely important to indicate areas where active surveillance should be performed [1, 35]. Thus, investments in public services are essential and urgent, with priority being given to the mobilization of financial resources for the control of *T. cruzi* vectors [1]. Vector control interventions must be carried out regularly, although in the last two decades they have progressively decreased due to changes in the technical and political priorities of Brazil [1, 29].

## Conclusions

The epidemiological scenario presented here suggests a real risk of vector transmission of *T. cruzi* to humans and domestic animals that inhabit the investigated area, signaled by

entomological indicators that indicate an increase in triatomine colonies in the house, in addition to chemical control of the populations of vectors of *T. cruzi* have been discontinued. Hence, several species of triatomines, mainly *T. brasiliensis* and *T. pseudomaculata*, naturally infected by the parasite, has been identified in intradomiciliary and peridomiciliary ecotopes, highlighting the need for more effective public policies to control of these vectors.

## Supporting information

**S1 File. Risk of vectorial transmission.**
(PDF)

**S2 File. Housing units investigated.**
(PDF)

**S3 File. Housing units infested.**
(PDF)

**S4 File. Housing units sprayed.**
(PDF)

**S5 File. Housing units colonized.**
(PDF)

## Acknowledgments

The authors thank the State Secretariat of Public Health of Rio Grande do Norte, represented by the authorities and health agents of the Municipal Secretariats, for their indispensable support in field activities and for the provision of data during the development of this research. We thank Dr. Luke Baton for revising the manuscript.

## Author Contributions

**Conceptualization:** George Harisson Felinto Sampaio, Andressa Noronha Barbosa da Silva, Paulo Marcos da Matta Guedes, Antonia Claudia Jácome da Câmara, Lúcia Maria da Cunha Galvão.

**Data curation:** Nathan Ravi Medeiros Honorato, Lúcia Maria Abrantes Aguiar, Carlos Ramon do Nascimento Brito.

**Formal analysis:** George Harisson Felinto Sampaio, Andressa Noronha Barbosa da Silva, Christiane Carlos Araújo de Negreiros, Nathan Ravi Medeiros Honorato, Rand Randall Martins, Letícia Mikardya Lima Sales.

**Investigation:** George Harisson Felinto Sampaio.

**Methodology:** George Harisson Felinto Sampaio, Andressa Noronha Barbosa da Silva, Christiane Carlos Araújo de Negreiros, Nathan Ravi Medeiros Honorato, Rand Randall Martins, Letícia Mikardya Lima Sales.

**Resources:** Antonia Claudia Jácome da Câmara, Lúcia Maria da Cunha Galvão.

**Supervision:** George Harisson Felinto Sampaio, Andressa Noronha Barbosa da Silva, Carlos Ramon do Nascimento Brito, Antonia Claudia Jácome da Câmara, Lúcia Maria da Cunha Galvão.

**Validation:** Andressa Noronha Barbosa da Silva, Nathan Ravi Medeiros Honorato, Carlos Ramon do Nascimento Brito, Paulo Marcos da Matta Guedes, Lúcia Maria da Cunha Galvão.

**Visualization:** Andressa Noronha Barbosa da Silva, Christiane Carlos Araújo de Negreiros, Nathan Ravi Medeiros Honorato.

**Writing – original draft:** George Harisson Felinto Sampaio, Andressa Noronha Barbosa da Silva, Christiane Carlos Araújo de Negreiros, Nathan Ravi Medeiros Honorato, Rand Randall Martins, Carlos Ramon do Nascimento Brito, Paulo Marcos da Matta Guedes, Lúcia Maria da Cunha Galvão.

**Writing – review & editing:** George Harisson Felinto Sampaio, Andressa Noronha Barbosa da Silva, Christiane Carlos Araújo de Negreiros, Nathan Ravi Medeiros Honorato, Rand Randall Martins, Lúcia Maria Abrantes Aguiar, Letícia Mikardya Lima Sales, Carlos Ramon do Nascimento Brito, Paulo Marcos da Matta Guedes, Antonia Claudia Jácome da Câmara, Lúcia Maria da Cunha Galvão.

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
