## [Decision Letter · Decision Letter 0]

6 Oct 2022

PONE-D-22-24351Temporal assessment of entomological surveillance and control of Trypanosoma cruzi vectors in an endemic area of northeastern Brazil.PLOS ONE

Dear Dr. Sampaio,

Thank you for submitting your manuscript to PLOS ONE. After careful consideration, we feel that it has merit but does not fully meet PLOS ONE’s publication criteria as it currently stands. Therefore, we invite you to submit a revised version of the manuscript that addresses the points raised during the review process. Both reviewers identified merits in the manuscript to be published in PLOS ONE, but highlighted some important changes to be done before its acceptance, especially concerning the alignment of Objectives, Methods, Results, Discussion and Conclusion. We will be delighted to receive the revised manuscript soon.

We look forward to receiving your revised manuscript.

Kind regards,

André Luiz Rodrigues Roque

Academic Editor

PLOS ONE

Journal Requirements:

"This work has not been submitted elsewhere for publication and will not be until we receive your reply regarding publication in your Journal. And also, if accepted, it will not be published elsewhere in the same form, in English or in any other language, including electronically without the written consent of the copyright-holder."

5. We note that Figure 1, S1, S2, S3 and S4 in your submission contain [map/satellite] images which may be copyrighted. All PLOS content is published under the Creative Commons Attribution License (CC BY 4.0), which means that the manuscript, images, and Supporting Information files will be freely available online, and any third party is permitted to access, download, copy, distribute, and use these materials in any way, even commercially, with proper attribution. For these reasons, we cannot publish previously copyrighted maps or satellite images created using proprietary data, such as Google software (Google Maps, Street View, and Earth). For more information, see our copyright guidelines: http://journals.plos.org/plosone/s/licenses-and-copyright.

a. You may seek permission from the original copyright holder of Figure 1, S1, S2, S3 and S4 to publish the content specifically under the CC BY 4.0 license.  

6. Please include your tables as part of your main manuscript and remove the individual files. Please note that supplementary tables (should remain/ be uploaded) as separate "supporting information" files

Reviewers' comments:

Reviewer's Responses to Questions

**Comments to the Author**

1. Is the manuscript technically sound, and do the data support the conclusions?

Reviewer #1: Partly

Reviewer #2: Yes

2. Has the statistical analysis been performed appropriately and rigorously? 

Reviewer #1: Yes

Reviewer #2: Yes

3. Have the authors made all data underlying the findings in their manuscript fully available?

Reviewer #1: Yes

Reviewer #2: Yes

4. Is the manuscript presented in an intelligible fashion and written in standard English?

Reviewer #1: Yes

Reviewer #2: Yes

5. Review Comments to the Author

Reviewer #1: The presented article describes the impact of triatomine surveillance and control measures between 2005 and 2015 in an endemic area for Chagas disease in the state of Rio Grande do Norte, Brazil.

This is an interesting approach and brings relevant information about vector control of triatomines in Agreste mesoregion, endemic area of Chagas disease, of the Rio Grande do Norte and describes changes in entomological surveillance metrics, used to monitor T. cruzi vectors in Brazil.

The methods are adequate in part to meet the proposed objectives, and the results and discussion are well described, needing some review and organization of the presented ideas. The conclusion section must be reviewed to align to presented objectives.

The manuscript has merits for be published in PLOS ONE, however, a few changes and corrections must be made before its acceptance.

General Comments

1 – Consider using the nomenclature “T.cruzi-like” instead of “T.cruzi” when referring to the diagnosis observed in the parasitological examination. Through the fresh parasitological examination, it is not possible to specifically distinguish T. cruzi from other species of the Trypanosoma genus, or even other flagellated protozoa of the Trypanosomatidae Family. Only molecular diagnosis can differentiate the species of the genus, and sometimes with a great difficulty.

2 – Consider review the objective: “Therefore, this study evaluated the impact of triatomine surveillance and control measures between 2005 and 2015 in an endemic area for ChD in the state of Rio Grande do Norte, Brazil.”

Here some arguments: your study evaluated the impact of triatomine surveillance on/over what? I mean, your objective was to describe the presented data or was to evaluate the impact over some specific determinant over the time? Please, consider reviewing the objectives to guide you along the Results, Discussion and Conclusion section. Also, these sections need to be better align with your objectives.

3 – Review the taxonomic rules of the scientific nomenclature. Review the abbreviations of the Triatoma genus and other genera along with all of the text.

4 - Consider amplifying the impact of your results. For example, what is the impact of the data presented for Brazil and for the world?

Methods

1 – Describe your study design. Prospective, retrospective? Align the study methods to your scientific questions and main reviewed objectives.

Statistical Analysis

2 – Describe the software or packages used to analyze and geoprocessing the data.

3 – Do you describe β coefficient in the Results section, but don’t describe this analysis in the Methods section.

Results

1 - In the first paragraph of the results, between lines 133 and 137, it describes information regarding the method session. Consider relocating the information to the correct section.

Discussion

1 - Consider reviewing all this section, better organize the ideas to maintain the same sequence of information described at the results section, and align to your objectives.

Conclusions

1 - Consider review this section to align with the proposed objective.

References

1 - Consider review, some references do not present the DOI, some are not adequately formatted.

Specific Comments

Line 6 – “Active entomological” – these data also include captured triatomines reported by passive surveillance through PITs? (Triatomine Information Post).

Line 25 – First taxonomic description of Trypanosoma cruzi. Consider include the complete description of the taxa - Trypanosoma cruzi (Chagas, 1909) – as presented for the triatomine names. Also, to the other taxonomic entities.

Line 45 – Consider reviewing the argument presented as evidence of active transmission of T. cruzi. A study (https://pubmed.ncbi.nlm.nih.gov/21249297) showed that about 3.7% of mothers were infected with T. cruzi and 2.5% of their newborn children acquired the parasite congenitally. (mother and child). Other values indicate even higher numbers “Argentina (6%), in 292 of 7,086 in Bolivia (4.1%) and in 115 of 2,691 in Paraguay (4.3%)” (https://doi.org/10.1590/0074-02760140405). The data presented in your manuscript informs “...1 out of 1,750 samples analyzed from the state of RN was positive for anti-T. crossed antibodies…”. This indicates a prevalence of 0.058%, much lower than what was observed in transmission without the presence of the vector. Despite of this, these are epidemiologically distinct scenarios, so my suggestion to the authors is that they consider better developing the presented rationale of this paragraph.

Line 50-52 – Since triatomine species have already been cited, consider writing the cited species in abbreviated form. Additionally, the full name of the species P. megistus has the wrong descriptor (Neiva and Pinto, 1923).

Line 71 – instead of “better control with ChD” consider “better control of ChD transmission.”

Line 100 – Please, consider better describing “exhaustively.”

Line 105 – Consider T.cruzi-like when referring to parasitological test.

Line 158-159 – Duplicated word - “index.”

Line 165-167 – Once this triatomines species is already presented in the text, consider abbreviating their genus. Triatoma pseudomaculata as T. pseudomaculata.

Also, please observe Triatoma Brasiliensis.

Lines 169-170 – Once these triatomine species are already presented in the text, consider abbreviating their genus.

Lines 187-190 – Consider review the sentence: “The risk of transmission of T. cruzi persists due to the existence of autochthonous triatomine species with colonization potential and the persistence of residual foci of Triatoma infestans in both sylvatic [39] and domestic environments [40,41].” Here some arguments: This specie was not part of your sample, there are no longer colonies of this specie in Brazil, Triatoma infestans in sylvatic environments does not occur in Brazil.

Lines 198 – Consider review the sentence: “The infested areas without residual spraying become a favorable habitat for the proliferation of triatomines, with abundant vertebrate hosts for blood-feeding because of the availability of humans and various species of domestic and peridomestic animals, in addition to resting sites and hiding places in the HU. Thus, spraying the adjacent surrounding areas prevents large-scale spatial displacement from infested areas of triatomines in search of bloodmeals.”

Line 247-249 – Consider review the sentence: “The main objective of ChD control programs is to eradicate domestic colonies of triatomines and maintain continuous surveillance against new-colonization” by “The main objective of ChD vector control programs is to eradicate domestic colonies of triatomines and maintain continuous surveillance of T. cruzi vectors.”

Line 253 – Consider review the sentence: “In low-risk municipalities” by “In municipalities classified as low-risk for T. cruzi transmission”

Line 277 – Consider review the sentence: “have been reported” by “have been reported previously.”

Line 288 – Consider review the sentence: “but a specificity of 100% for

T. cruzi.” Only molecular methods can differentiate T. cruzi from others Trypanosomatidae (Family). May I ask you, which were the taxonomic markers do you used to differentiate T. cruzi from T. donovanni, at parasitological examination, for example?

Figures

Figure 1. The presented figure needs to be intensively reviewed. It’s not necessary to keep the author of the figure at the figure. Two legends of color degrade are presented in the figure, but it’s not clear to which map it belongs, please review the figure. Consider insert legends (A, B and C) to each tree maps presented at the Figure 1. Also, better describe this information at Figure 1 description.

Reviewer #2: General comments

The manuscript (MS) deals with an analysis of data from the health secretariats on the work carried out by them in vector control. The period covered by the analyzes covers 10 years (2005-2015). The MS is very well written and the data were adequately treated statistically. The authors concluded that the coverage of control measures has been decreasing in recent years, which is very worrying. Therefore, bringing this data to the public is important and so I am in favor of publishing this communication. Below I present a few suggestions and corrections for MS improvement.

Minor comments

Abstract

Ln 8-10: “The quantitative9 analysis of housing units surveyed for entomological indicators was performed by linear regression of random effects (p < 0.05).”

Please, let clear which comparison was significant

“and control of vectors”

This piece is in bold. It needs to be in regular font.

Along the whole text

Species names should be written in full at first, but for later mentions, the genus name must be abbreviated. E.g. “Triatoma brasiliensis (= “T. brasiliensis”).

Ln 220. I assume the word “susing” is in fact “using”

Ln 245-246: “Interventions such as improving housing conditions and organizing the peridomicile [1,52], as well as keeping dogs, cats and rodents out of human sleeping areas”

This recommendation is realistic only for rodents. Please, re-thing its necessity.

Ln 291: “291 for diagnosis, showed natural infection rates of up to 79% in Triatoma brasiliensis [20].”

In fact, Lilioso and colleagues found up to 100% of natural infection (see Fig 3). What is worse, this population was collected aside from the sugar cane mill responsible for the outbreak in Marcelino Vieira.

CEA

6. PLOS authors have the option to publish the peer review history of their article (what does this mean?). If published, this will include your full peer review and any attached files.

Reviewer #1: No

Reviewer #2: **Yes: **Carlos Eduardo Almeida

---

## [Author Response · Author response to Decision Letter 0]

10 Mar 2023

General Comments

1 – Consider using the nomenclature “T.cruzi-like” instead of “T.cruzi” when referring to the diagnosis observed in the parasitological examination. Through the fresh parasitological examination, it is not possible to specifically distinguish T. cruzi from other species of the Trypanosoma genus, or even other flagellated protozoa of the Trypanosomatidae Family. Only molecular diagnosis can differentiate the species of the genus, and sometimes with a great difficulty.

1 - Answer: The term the nomenclature “T.cruzi-like” was used as suggested. Please see in the text of the manuscript highlighted yellow.

2 – Consider review the objective: “Therefore, this study evaluated the impact of triatomine surveillance and control measures between 2005 and 2015 in an endemic area for ChD in the state of Rio Grande do Norte, Brazil.” 

Here some arguments: your study evaluated the impact of triatomine surveillance on/over what? I mean, your objective was to describe the presented data or was to evaluate the impact over some specific determinant over the time? Please, consider reviewing the objectives to guide you along the Results, Discussion and Conclusion section. Also, these sections need to be better align with your objectives.

2 - Answer: The objectives have been revised and edited as suggested. . Please see in the text of the manuscript highlighted yellow. The new objectives has been aligned with the results, discussion and conclusion sections.

3 – Review the taxonomic rules of the scientific nomenclature. Review the abbreviations of the Triatoma genus and other genera along with all of the text. 

3 - Answer: The taxonomic rules of scientific nomenclature have been revised, particulary the abbreviations of the genus Triatoma and other genera. Please see in the text of the manuscript highlighted yellow. 

Methods

1 – Describe your study design. Prospective, retrospective? Align the study methods to your scientific questions and main reviewed objectives.

1 - Answer: The study design was described and aligned with the methods, its scientific questions and the main objectives reviewed as suggested. Please see in the text of the manuscript highlighted yellow.

Statistical Analysis

2 – Describe the software or packages used to analyze and geoprocessing the data.

2 - Answer: The software used to analyze and geoprocess the data was described in the methods section. Please see in the text of the manuscript highlighted yellow.

3 – Do you describe β coefficient in the Results section, but don’t describe this analysis in the Methods section.

3 - Answer: The β coefficient was described in the methods section. Please see in the text of the manuscript highlighted yellow.

4 – Consider amplifying the impact of your results. For example, what is the impact of the data presented for Brazil and for the world?

4 - Answer: The reviewer`s suggestion was accepted. Please see in the text of the manuscript highlighted yellow.

Results

1 – In the first paragraph of the results, between lines 133 and 137, it describes information regarding the method session. Consider relocating the information to the correct section.

1 - Answer: In the first paragraph the results that described information regarding the method session were relocated in the correct section as suggested. Please see in the text of the manuscript highlighted yellow.

Discussion 

1 – Consider reviewing all this section, better organize the ideas to maintain the same sequence of information described at the results section, and align to your objectives.

1 - Answer: The reviewer`s suggestion was accepted and the following of information and ideas presented in the objective were aligned with the results and discussion sections as suggested. Please see in the text of the manuscript highlighted yellow.

Conclusions

1 – Consider review this section to align with the proposed objective.

1 - Answer: This section was aligned with the proposed new objective as suggested. Please see in the text of the manuscript highlighted yellow.

References

1 – Consider review, some references do not present the DOI, some are not adequately formatted.

1 - Answer: References have been reviewed and formatted properly as suggested.

Specific Comments

Line 6 – “Active entomological” – these data also include captured triatomines reported by passive surveillance through PITs? (Triatomine Information Post).

Answer: No, these data refer only to Triatomines captured by endemic agents in households.

Line 25 – First taxonomic description of Trypanosoma cruzi. Consider include the complete description of the taxa - Trypanosoma cruzi (Chagas, 1909) – as presented for the triatomine names. Also, to the other taxonomic entities.

Answer: The first taxonomic description of Trypanosoma cruzi and other taxonomic entities were included as suggested. Please see in the text of the manuscript highlighted yellow.

Line 45 – Consider reviewing the argument presented as evidence of active transmission of T. cruzi. A study (https://pubmed.ncbi.nlm.nih.gov/21249297) showed that about 3.7% of mothers were infected with T. cruzi and 2.5% of their newborn children acquired the parasite congenitally. (mother and child). Other values indicate even higher numbers “Argentina (6%), in 292 of 7,086 in Bolivia (4.1%) and in 115 of 2,691 in Paraguay (4.3%)” (https://doi.org/10.1590/0074-02760140405). The data presented in your manuscript informs “...1 out of 1,750 samples analyzed from the state of RN was positive for anti-T. crossed antibodies…”. This indicates a prevalence of 0.058%, much lower than what was observed in transmission without the presence of the vector. Despite of this, these are epidemiologically distinct scenarios, so my suggestion to the authors is that they consider better developing the presented rationale of this paragraph.

Answer: The argument presented as evidence of active transmission of T. cruzi in the studied region, regarding seropositivity in children, was presented in another order of importance in the text as suggested. Please see in the text of the manuscript highlighted yellow.

Line 50-52 – Since triatomine species have already been cited, consider writing the cited species in abbreviated form. Additionally, the full name of the species P. megistus has the wrong descriptor (Neiva and Pinto, 1923).

Answer: Triatomine species were abbreviated after being cited. In addition, the full name of the P. megistus species has been corrected. Please see in the text of the manuscript highlighted yellow.

Line 71 – instead of “better control with ChD” consider “better control of ChD transmission.”

Answer: The reviewer`s suggestion was accepted. Please see in the text of the manuscript highlighted yellow.

Line 100 – Please, consider better describing “exhaustively.”

Answer: The word “exhaustively” has been removed from the text. Please see in the text of the manuscript highlighted yellow.

Line 105 – Consider T. cruzi-like when referring to parasitological test.

Answer: The reviewer`s suggestion was accepted. Please see in the text of the manuscript highlighted yellow.

Line 158-159 – Duplicated word - “index.”

Answer: Duplicated word - “index” has been removed from the text. Please see in the text of the manuscript highlighted yellow.

Line 165-167 – Once this triatomines species is already presented in the text, consider abbreviating their genus. Triatoma pseudomaculata as T. pseudomaculata.

Also, please observe Triatoma Brasiliensis.

Answer: The genus was later abbreviated and Triatoma Brasiliensis” was corrected.

Lines 169-170 – Once these triatomine species are already presented in the text, consider abbreviating their genus.

Answer: Your suggestion was accepted. Please see in the text of the manuscript highlighted yellow.

Lines 187-190 – Consider review the sentence: “The risk of transmission of T. cruzi persists due to the existence of autochthonous triatomine species with colonization potential and the persistence of residual foci of Triatoma infestans in both sylvatic [39] and domestic environments [40,41].” Here some arguments: This specie was not part of your sample, there are no longer colonies of this specie in Brazil, Triatoma infestans in sylvatic environments does not occur in Brazil.

Answer: The sentence regarding Triatoma infestans has been removed as suggested. Please see in the text of the manuscript highlighted yellow.

Lines 198 – Consider review the sentence: “The infested areas without residual spraying become a favorable habitat for the proliferation of triatomines, with abundant vertebrate hosts for blood-feeding because of the availability of humans and various species of domestic and peridomestic animals, in addition to resting sites and hiding places in the HU. Thus, spraying the adjacent surrounding areas prevents large-scale spatial displacement from infested areas of triatomines in search of bloodmeals.”

Answer: The sentence has been revised and rewritten more linearly. Please see in the text of the manuscript highlighted yellow.

Line 247-249 – Consider review the sentence: “The main objective of ChD control programs is to eradicate domestic colonies of triatomines and maintain continuous surveillance against new-colonization” by “The main objective of ChD vector control programs is to eradicate domestic colonies of triatomines and maintain continuous surveillance of T. cruzi vectors.” 

Answer: Your suggestion was accepted. Please see in the text of the manuscript highlighted yellow.

Line 253 – Consider review the sentence: “In low-risk municipalities” by “In municipalities classified as low-risk for T. cruzi transmission”

Answer: The sentence was revised as suggested. Please see in the text of the manuscript highlighted yellow.

Line 277 – Consider review the sentence: “have been reported” by “have been reported previously.” 

Answer: The sentence was revised as suggested. Please see in the text of the manuscript highlighted yellow.

Line 288 – Consider review the sentence: “but a specificity of 100% forT. cruzi.” Only molecular methods can differentiate T. cruzi from others Trypanosomatidae (Family). May I ask you, which were the taxonomic markers do you used to differentiate T. cruzi from T. donovanni, at parasitological examination, for example?

Answer: The sentence was revised as suggested. Please see in the text of the manuscript highlighted yellow.

Figures

Figure 1. The presented figure needs to be intensively reviewed. It’s not necessary to keep the author of the figure at the figure. Two legends of color degrade are presented in the figure, but it’s not clear to which map it belongs, please review the figure. Consider insert legends (A, B and C) to each tree maps presented at the Figure 1. Also, better describe this information at Figure 1 description.

Answer: Figure 1 was reviewed as suggested. Please see the new Figure 1.

---

## [Editor Report · Decision Letter 1]

10 Apr 2023

PONE-D-22-24351R1Temporal assessment of entomological surveillance of Trypanosoma cruzi vectors in an endemic area of northeastern BrazilPLOS ONE

Dear Dr. Sampaio,

This version was not reviewed by me. Authors have to answer the comments of the second reviewer too, modifying the manuscript, when applicable. And then, resubmit the R1 version of the manuscript with comments to both reviewers.

We look forward to receiving your revised manuscript.

Kind regards,

André Luiz Rodrigues Roque

Academic Editor

PLOS ONE
---

## [Author Response · Author response to Decision Letter 1]

11 Apr 2023

Specific Comments

Line 6 – “Active entomological” – these data also include captured triatomines reported by passive surveillance through PITs? (Triatomine Information Post).

Answer: No, these data refer only to Triatomines captured by endemic agents in households.

Line 25 – First taxonomic description of Trypanosoma cruzi. Consider include the complete description of the taxa - Trypanosoma cruzi (Chagas, 1909) – as presented for the triatomine names. Also, to the other taxonomic entities.

Answer: The first taxonomic description of Trypanosoma cruzi and other taxonomic entities were included as suggested. Please see in the text of the manuscript highlighted yellow.

Line 45 – Consider reviewing the argument presented as evidence of active transmission of T. cruzi. A study (https://pubmed.ncbi.nlm.nih.gov/21249297) showed that about 3.7% of mothers were infected with T. cruzi and 2.5% of their newborn children acquired the parasite congenitally. (mother and child). Other values indicate even higher numbers “Argentina (6%), in 292 of 7,086 in Bolivia (4.1%) and in 115 of 2,691 in Paraguay (4.3%)” (https://doi.org/10.1590/0074-02760140405). The data presented in your manuscript informs “...1 out of 1,750 samples analyzed from the state of RN was positive for anti-T. crossed antibodies…”. This indicates a prevalence of 0.058%, much lower than what was observed in transmission without the presence of the vector. Despite of this, these are epidemiologically distinct scenarios, so my suggestion to the authors is that they consider better developing the presented rationale of this paragraph.

Answer: The argument presented as evidence of active transmission of T. cruzi in the studied region, regarding seropositivity in children, was presented in another order of importance in the text as suggested. Please see in the text of the manuscript highlighted yellow.

Line 50-52 – Since triatomine species have already been cited, consider writing the cited species in abbreviated form. Additionally, the full name of the species P. megistus has the wrong descriptor (Neiva and Pinto, 1923).

Answer: Triatomine species were abbreviated after being cited. In addition, the full name of the P. megistus species has been corrected. Please see in the text of the manuscript highlighted yellow.

Line 71 – instead of “better control with ChD” consider “better control of ChD transmission.”

Answer: The reviewer`s suggestion was accepted. Please see in the text of the manuscript highlighted yellow.

Line 100 – Please, consider better describing “exhaustively.”

Answer: The word “exhaustively” has been removed from the text. Please see in the text of the manuscript highlighted yellow.

Line 105 – Consider T. cruzi-like when referring to parasitological test.

Answer: The reviewer`s suggestion was accepted. Please see in the text of the manuscript highlighted yellow.

Line 158-159 – Duplicated word - “index.”

Answer: Duplicated word - “index” has been removed from the text. Please see in the text of the manuscript highlighted yellow.

Line 165-167 – Once this triatomines species is already presented in the text, consider abbreviating their genus. Triatoma pseudomaculata as T. pseudomaculata.

Also, please observe Triatoma Brasiliensis.

Answer: The genus was later abbreviated and Triatoma Brasiliensis” was corrected.

Lines 169-170 – Once these triatomine species are already presented in the text, consider abbreviating their genus.

Answer: Your suggestion was accepted. Please see in the text of the manuscript highlighted yellow.

Lines 187-190 – Consider review the sentence: “The risk of transmission of T. cruzi persists due to the existence of autochthonous triatomine species with colonization potential and the persistence of residual foci of Triatoma infestans in both sylvatic [39] and domestic environments [40,41].” Here some arguments: This specie was not part of your sample, there are no longer colonies of this specie in Brazil, Triatoma infestans in sylvatic environments does not occur in Brazil.

Answer: The sentence regarding Triatoma infestans has been removed as suggested. Please see in the text of the manuscript highlighted yellow.

Lines 198 – Consider review the sentence: “The infested areas without residual spraying become a favorable habitat for the proliferation of triatomines, with abundant vertebrate hosts for blood-feeding because of the availability of humans and various species of domestic and peridomestic animals, in addition to resting sites and hiding places in the HU. Thus, spraying the adjacent surrounding areas prevents large-scale spatial displacement from infested areas of triatomines in search of bloodmeals.”

Answer: The sentence has been revised and rewritten more linearly. Please see in the text of the manuscript highlighted yellow.

Line 247-249 – Consider review the sentence: “The main objective of ChD control programs is to eradicate domestic colonies of triatomines and maintain continuous surveillance against new-colonization” by “The main objective of ChD vector control programs is to eradicate domestic colonies of triatomines and maintain continuous surveillance of T. cruzi vectors.” 

Answer: Your suggestion was accepted. Please see in the text of the manuscript highlighted yellow.

Line 253 – Consider review the sentence: “In low-risk municipalities” by “In municipalities classified as low-risk for T. cruzi transmission”

Answer: The sentence was revised as suggested. Please see in the text of the manuscript highlighted yellow.

Line 277 – Consider review the sentence: “have been reported” by “have been reported previously.” 

Answer: The sentence was revised as suggested. Please see in the text of the manuscript highlighted yellow.

Line 288 – Consider review the sentence: “but a specificity of 100% forT. cruzi.” Only molecular methods can differentiate T. cruzi from others Trypanosomatidae (Family). May I ask you, which were the taxonomic markers do you used to differentiate T. cruzi from T. donovanni, at parasitological examination, for example?

Answer: The sentence was revised as suggested. Please see in the text of the manuscript highlighted yellow.

Figures

Figure 1. The presented figure needs to be intensively reviewed. It’s not necessary to keep the author of the figure at the figure. Two legends of color degrade are presented in the figure, but it’s not clear to which map it belongs, please review the figure. Consider insert legends (A, B and C) to each tree maps presented at the Figure 1. Also, better describe this information at Figure 1 description.

Answer: Figure 1 was reviewed as suggested. Please see the new Figure 1.

---

## [Editor Report · Decision Letter 2]

14 Apr 2023

PONE-D-22-24351R2Temporal assessment of entomological surveillance of Trypanosoma cruzi vectors in an endemic area of northeastern BrazilPLOS ONE

Dear Dr. Sampaio,

Thank you for submitting your manuscript to PLOS ONE. After careful consideration, we feel that it has merit but does not fully meet PLOS ONE’s publication criteria as it currently stands. Therefore, we invite you to submit a revised version of the manuscript that addresses the points raised during the review process.

Once again, authors submitted the response letter and modified manuscript without considering the comments by reviewer 2. Maybe, authors did not find the comments in the system, then, I'm pasting below the comments of Reviewer 2 that have to be answered and modified, when applicable.

General comments

The manuscript (MS) deals with an analysis of data from the health secretariats on the work carried out by them in vector control. The period covered by the analyzes covers 10 years (2005-2015). The MS is very well written and the data were adequately treated statistically. The authors concluded that the coverage of control measures has been decreasing in recent years, which is very worrying. Therefore, bringing this data to the public is important and so I am in favor of publishing this communication. Below I present a few suggestions and corrections for MS improvement.

Minor comments

Abstract

Ln 8-10: “The quantitative9 analysis of housing units surveyed for entomological indicators was performed by linear regression of random effects (p < 0.05).”

Please, let clear which comparison was significant

“and control of vectors”

This piece is in bold. It needs to be in regular font.

Along the whole text

Species names should be written in full at first, but for later mentions, the genus name must be abbreviated. E.g. “Triatoma brasiliensis (= “T. brasiliensis”).

Ln 220. I assume the word “susing” is in fact “using”

Ln 245-246: “Interventions such as improving housing conditions and organizing the peridomicile [1,52], as well as keeping dogs, cats and rodents out of human sleeping areas”

This recommendation is realistic only for rodents. Please, re-thing its necessity.

Ln 291: “291 for diagnosis, showed natural infection rates of up to 79% in Triatoma brasiliensis [20].”

In fact, Lilioso and colleagues found up to 100% of natural infection (see Fig 3). What is worse, this population was collected aside from the sugar cane mill responsible for the outbreak in Marcelino Vieira.

We look forward to receiving your revised manuscript.

Kind regards,

André Luiz Rodrigues Roque

Academic Editor

PLOS ONE
---

## [Author Response · Author response to Decision Letter 2]

27 Apr 2023

Reviewer 1

Specific Comments

Line 6 – “Active entomological” – these data also include captured triatomines reported by passive surveillance through PITs? (Triatomine Information Post).

Answer: No, these data refer only to Triatomines captured by endemic agents in households.

Line 25 – First taxonomic description of Trypanosoma cruzi. Consider include the complete description of the taxa - Trypanosoma cruzi (Chagas, 1909) – as presented for the triatomine names. Also, to the other taxonomic entities.

Answer: The first taxonomic description of Trypanosoma cruzi and other taxonomic entities were included as suggested. Please see in the text of the manuscript highlighted yellow.

Line 45 – Consider reviewing the argument presented as evidence of active transmission of T. cruzi. A study (https://pubmed.ncbi.nlm.nih.gov/21249297) showed that about 3.7% of mothers were infected with T. cruzi and 2.5% of their newborn children acquired the parasite congenitally. (mother and child). Other values indicate even higher numbers “Argentina (6%), in 292 of 7,086 in Bolivia (4.1%) and in 115 of 2,691 in Paraguay (4.3%)” (https://doi.org/10.1590/0074-02760140405). The data presented in your manuscript informs “...1 out of 1,750 samples analyzed from the state of RN was positive for anti-T. crossed antibodies…”. This indicates a prevalence of 0.058%, much lower than what was observed in transmission without the presence of the vector. Despite of this, these are epidemiologically distinct scenarios, so my suggestion to the authors is that they consider better developing the presented rationale of this paragraph.

Answer: The argument presented as evidence of active transmission of T. cruzi in the studied region, regarding seropositivity in children, was presented in another order of importance in the text as suggested. Please see in the text of the manuscript highlighted yellow.

Line 50-52 – Since triatomine species have already been cited, consider writing the cited species in abbreviated form. Additionally, the full name of the species P. megistus has the wrong descriptor (Neiva and Pinto, 1923).

Answer: Triatomine species were abbreviated after being cited. In addition, the full name of the P. megistus species has been corrected. Please see in the text of the manuscript highlighted yellow.

Line 71 – instead of “better control with ChD” consider “better control of ChD transmission.”

Answer: The reviewer`s suggestion was accepted. Please see in the text of the manuscript highlighted yellow.

Line 100 – Please, consider better describing “exhaustively.”

Answer: The word “exhaustively” has been removed from the text. Please see in the text of the manuscript highlighted yellow.

Line 105 – Consider T. cruzi-like when referring to parasitological test.

Answer: The reviewer`s suggestion was accepted. Please see in the text of the manuscript highlighted yellow.

Line 158-159 – Duplicated word - “index.”

Answer: Duplicated word - “index” has been removed from the text. Please see in the text of the manuscript highlighted yellow.

Line 165-167 – Once this triatomines species is already presented in the text, consider abbreviating their genus. Triatoma pseudomaculata as T. pseudomaculata.

Also, please observe Triatoma Brasiliensis.

Answer: The genus was later abbreviated and Triatoma Brasiliensis” was corrected.

Lines 169-170 – Once these triatomine species are already presented in the text, consider abbreviating their genus.

Answer: Your suggestion was accepted. Please see in the text of the manuscript highlighted yellow.

Lines 187-190 – Consider review the sentence: “The risk of transmission of T. cruzi persists due to the existence of autochthonous triatomine species with colonization potential and the persistence of residual foci of Triatoma infestans in both sylvatic [39] and domestic environments [40,41].” Here some arguments: This specie was not part of your sample, there are no longer colonies of this specie in Brazil, Triatoma infestans in sylvatic environments does not occur in Brazil.

Answer: The sentence regarding Triatoma infestans has been removed as suggested. Please see in the text of the manuscript highlighted yellow.

Lines 198 – Consider review the sentence: “The infested areas without residual spraying become a favorable habitat for the proliferation of triatomines, with abundant vertebrate hosts for blood-feeding because of the availability of humans and various species of domestic and peridomestic animals, in addition to resting sites and hiding places in the HU. Thus, spraying the adjacent surrounding areas prevents large-scale spatial displacement from infested areas of triatomines in search of bloodmeals.”

Answer: The sentence has been revised and rewritten more linearly. Please see in the text of the manuscript highlighted yellow.

Line 247-249 – Consider review the sentence: “The main objective of ChD control programs is to eradicate domestic colonies of triatomines and maintain continuous surveillance against new-colonization” by “The main objective of ChD vector control programs is to eradicate domestic colonies of triatomines and maintain continuous surveillance of T. cruzi vectors.” 

Answer: Your suggestion was accepted. Please see in the text of the manuscript highlighted yellow.

Line 253 – Consider review the sentence: “In low-risk municipalities” by “In municipalities classified as low-risk for T. cruzi transmission”

Answer: The sentence was revised as suggested. Please see in the text of the manuscript highlighted yellow.

Line 277 – Consider review the sentence: “have been reported” by “have been reported previously.” 

Answer: The sentence was revised as suggested. Please see in the text of the manuscript highlighted yellow.

Line 288 – Consider review the sentence: “but a specificity of 100% forT. cruzi.” Only molecular methods can differentiate T. cruzi from others Trypanosomatidae (Family). May I ask you, which were the taxonomic markers do you used to differentiate T. cruzi from T. donovanni, at parasitological examination, for example?

Answer: The sentence was revised as suggested. Please see in the text of the manuscript highlighted yellow.

Figures

Figure 1. The presented figure needs to be intensively reviewed. It’s not necessary to keep the author of the figure at the figure. Two legends of color degrade are presented in the figure, but it’s not clear to which map it belongs, please review the figure. Consider insert legends (A, B and C) to each tree maps presented at the Figure 1. Also, better describe this information at Figure 1 description.

Answer: Figure 1 was reviewed as suggested. Please see the new Figure 1.

Reviewer 2

Minor comments

Abstract

Ln 8-10: “The quantitative9 analysis of housing units surveyed for entomological indicators was performed by linear regression of random effects (p < 0.05).”

Please, let clear which comparison was significant. 

Answer: Suggested information has been added to the manuscript. Please see in the text of the manuscript highlighted yellow. 

“and control of vectors.” This piece is in bold. It needs to be in regular font.

Answer: Correction made to the manuscript. Please see in the text of the manuscript highlighted yellow.

Along the whole text Species names should be written in full at first, but for later mentions, the genus name must be abbreviated. E.g. “Triatoma brasiliensis (= “T. brasiliensis”).

 Answer: This correction was made throughout the manuscript. Please see in the text of the manuscript highlighted yellow

Ln 220. I assume the word “susing” is in fact “using”

Answer: This correction was made throughout the manuscript. Please see in the text of the manuscript highlighted yellow

Ln 245-246: “Interventions such as improving housing conditions and organizing the peridomicile [1,52], as well as keeping dogs, cats and rodents out of human sleeping areas” This recommendation is realistic only for rodents. Please, re-thing its necessity.

Answer: This correction was made throughout the manuscript. Please see in the text of the manuscript highlighted yellow

Ln 291: “291 for diagnosis, showed natural infection rates of up to 79% in Triatoma brasiliensis [20].” In fact, Lilioso and colleagues found up to 100% of natural infection (see Fig 3). What is worse, this population was collected aside from the sugar cane mill responsible for the outbreak in Marcelino Vieira.

Answer: This correction was made throughout the manuscript. Please see in the text of the manuscript highlighted yellow

---

## [Decision Letter · Decision Letter 3]

2 Jun 2023

Temporal assessment of entomological surveillance of Trypanosoma cruzi vectors in an endemic area of northeastern Brazil.

PONE-D-22-24351R3

Dear Dr. Sampaio,

We’re pleased to inform you that your manuscript has been judged scientifically suitable for publication and will be formally accepted for publication once it meets all outstanding technical requirements.

Kind regards,

André Luiz Rodrigues Roque

Academic Editor

PLOS ONE

Additional Editor Comments (optional):

Reviewers' comments:

Reviewer's Responses to Questions

**Comments to the Author**

1. If the authors have adequately addressed your comments raised in a previous round of review and you feel that this manuscript is now acceptable for publication, you may indicate that here to bypass the “Comments to the Author” section, enter your conflict of interest statement in the “Confidential to Editor” section, and submit your "Accept" recommendation.

Reviewer #1: All comments have been addressed

Reviewer #2: All comments have been addressed

2. Is the manuscript technically sound, and do the data support the conclusions?

Reviewer #1: Yes

Reviewer #2: Yes

3. Has the statistical analysis been performed appropriately and rigorously? 

Reviewer #1: Yes

Reviewer #2: Yes

4. Have the authors made all data underlying the findings in their manuscript fully available?

Reviewer #1: Yes

Reviewer #2: Yes

5. Is the manuscript presented in an intelligible fashion and written in standard English?

Reviewer #1: Yes

Reviewer #2: Yes

6. Review Comments to the Author

Reviewer #1: (No Response)

Reviewer #2: (No Response)

7. PLOS authors have the option to publish the peer review history of their article (what does this mean?). If published, this will include your full peer review and any attached files.

Reviewer #1: **Yes: **Gilmar Ribeiro-Jr.

Reviewer #2: No

---

## [Editor Report · Acceptance letter]

8 Jun 2023

PONE-D-22-24351R3 

Temporal assessment of entomological surveillance of *Trypanosoma cruzi* vectors in an endemic area of northeastern Brazil. 

Dear Dr. Sampaio:

I'm pleased to inform you that your manuscript has been deemed suitable for publication in PLOS ONE. Congratulations! Your manuscript is now with our production department. 

Kind regards, 

on behalf of

Dr. André Luiz Rodrigues Roque 

Academic Editor

PLOS ONE